# Maximizing regional biodiversity requires a mosaic of protection levels

**Nicolas Loiseau**[1,2,3]*, **Wilfried Thuiller**[2], **Rick D. Stuart-Smith**[4], **Vincent Devictor**[5], **Graham J. Edgar**[4], **Laure Velez**[1], **Joshua E. Cinner**[6], **Nicholas A. J. Graham**[7], **Julien Renaud**[2], **Andrew S. Hoey**[6], **Stephanie Manel**[8], **David Mouillot**[1,9]

**1** MARBEC, Univ Montpellier, CNRS, Ifremer, IRD, Montpellier, France, **2** Univ. Grenoble Alpes, Univ. Savoie Mont Blanc, CNRS, LECA, Laboratoire d'Ecologie Alpine, F-38000 Grenoble, France, **3** CEFE, Univ. Montpellier, CNRS, EPHE-PSL University, IRD, Univ Paul Valéry Montpellier 3, Montpellier, France, **4** Institute for Marine and Antarctic Studies, University of Tasmania, Hobart, Tasmania, Australia, **5** CNRS, ISEM, Université de Montpellier, IRD, EPHE, Montpellier, France, **6** ARC Centre of Excellence for Coral Reef Studies, James Cook University, Townsville, QLD, Australia, **7** Lancaster Environment Centre, Lancaster University, Lancaster, United Kingdom, **8** EPHE, PSL Research University, CNRS, UM, SupAgro, IRD, INRA, UMR 5175 CEFE, F-Montpellier, France, **9** Institut Universitaire de France, IUF, Paris, France

* nicolas.loiseau1@gmail.com

**Data Availability Statement:** All data used in this manuscript have already been published. Fish data are available on the Reef Life Survey web site (https://reeflifesurvey.com/reef-life-survey/survey-data/) and from Cinner et al. (2016)[34]. Bird data

## Abstract

Protected areas are the flagship management tools to secure biodiversity from anthropogenic impacts. However, the extent to which adjacent areas with distinct protection levels host different species numbers and compositions remains uncertain. Here, using reef fishes, European alpine plants, and North American birds, we show that the composition of species in adjacent Strictly Protected, Restricted, and Non-Protected areas is highly dissimilar, whereas the number of species is similar, after controlling for environmental conditions, sample size, and rarity. We find that between 12% and 15% of species are only recorded in Non-Protected areas, suggesting that a non-negligible part of regional biodiversity occurs where human activities are less regulated. For imperiled species, the proportion only recorded in Strictly Protected areas reaches 58% for fishes, 11% for birds, and 7% for plants, highlighting the fundamental and unique role of protected areas and their environmental conditions in biodiversity conservation.

## Introduction

Species diversity is changing at all spatial scales under ever-increasing habitat degradation, spread of non-native species, overexploitation of resources, and climate warming [1–4]. Ambitious conservation targets have been recently proposed to halt and even reverse the ongoing biodiversity erosion with the commitment to protect at least 30% of the global ocean and land by 2030 [5,6]. Yet, although a few studies indicate that species diversity tends to be higher inside than outside protected areas worldwide [7,8], the extent to which adjacent Strictly Protected, Restricted, and Non-Protected areas support different levels of species richness ($\alpha$-diversity) or different species compositions ($\beta$-diversity) remains unclear while being a key issue. Indeed, the stability of ecosystem functioning and the continuous delivery of ecosystem

(BBS surveys) are provided at https://www.pwrc.usgs.gov/bbs/. Alpine data are available from http://www.cbn-alpin.fr/ [46]. Official coverages of all existing protected areas in US and France (Protected Areas Database of the United States and The World Database on Protected Areas, WDPA) are respectively available at https://gapanalysis.usgs.gov/padus/ and https://www.protectedplanet.net/. Land cover in the US and France are respectively available at https://catalog.data.gov/dataset and http://www.theia-land.fr/.

**Funding:** Funding and support come from the European project RESERVEBENEFIT (European call BIODIVERSA3 2015-2016 call, SM). Additional support was provided by Australian Research Council [CE140100020, FT160100047] (JEC) & Australian Research Council [LP150100761] (GJE), the Ian Potter Foundation (GJE), the Royal Society University Research Fellowship (UF140691, NAJG), the ANR project Origin-Alps (ANR-16-CE93-0004,WT & JR) and from 'Investissement d'Avenir' grants managed by the ANR (Trajectories: ANR-15-IDEX-02; Montane: OSUG@2020: ANR-10-LAB-56, WT & JR). The funders had no role in study design, data collection and analysis, decision to publish, or preparation of the manuscript.

**Competing interests:** The authors have declared that no competing interests exist.

**Abbreviations:** BBS, Breeding Bird Survey; CBNA, National Alpine Botanical Conservatory; CR, Critically Endangered; dbRDA, distance-based redundancy analyses; DD, Data Deficient; EN, Endangered; IUCN, The International Union for Conservation of Nature; LC, Least Concern; NLCD, National Land Cover Database; NPA, Non-Protected Areas; NT, Near Threatened; RA, Restricted Areas; RLS, Reef Life Survey; SES, standardized effect sizes; SPA, Strictly Protected Areas; UVC, underwater visual census; VU, Vulnerable; WDPA, World Database on Protected Areas.

services at the regional scale is positively related to the number of species comprising the regional pool (γ-diversity) [9–11] which depends on both local or site diversity (α-diversity) and the dissimilarity in species composition between sites (ß-diversity) [12,13].

Historically, many protected areas have been created to conserve iconic landscapes and seascapes and to provide favorable environmental conditions (habitat and climate) for exploited and threatened species [14]. Such arbitrary choices may have contributed to a marked species dissimilarity between adjacent Protected and Non-Protected areas primarily due to habitat differences within a region. Beyond this potential habitat effect, some conservation-dependent species can be extirpated by human activities outside protected areas [3,15,16] which may also increase species dissimilarity between protection levels. So, on the balance, the relative influence of local environmental conditions and protection level on species composition is still largely unknown, while the partial influence of protection level when controlling for environmental conditions, sample size, and species occurrences remains to be investigated across taxa and biomes. Here, we make the hypothesis that protected areas host higher α-diversity than their non-protected counterparts with a nested pattern in ß-diversity where non-protected areas host only a subset of those species present in nearby protected areas (Fig 1). As an alternative, protected and non-protected may host different species inducing a mechanical increase in γ-diversity (Fig 1).

More specifically, we use 3 extensive and species-rich datasets across many regions (1,447 reef fish species in 44 protected areas, 3,532 plant species of the French Alps in 192 protected areas, and 639 North American bird species in 415 protected areas; see S1 Table) to compare

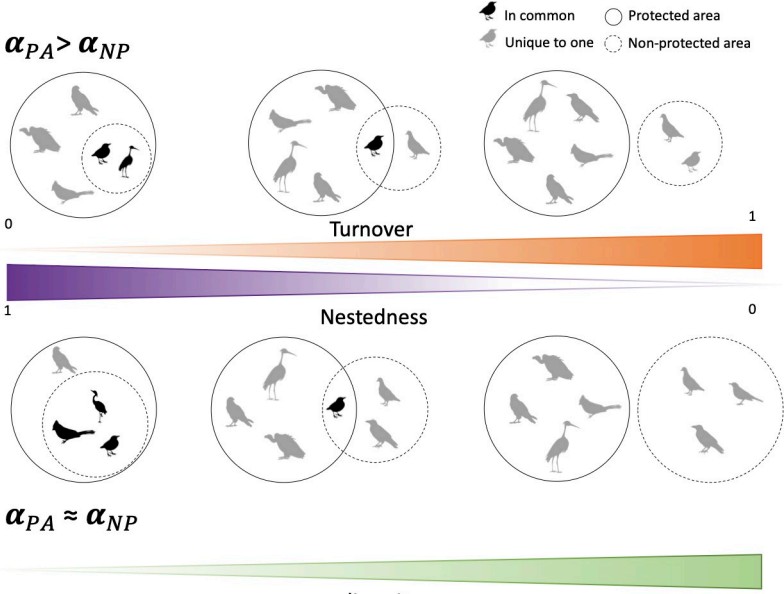

**Fig 1. Illustration of species dissimilarity and ß-diversity partitioning between 2 components—species turnover and species nestedness with a theoretical comparison between a protected and a non-protected area sharing some species.** This hypothetical example considers only 2 levels of protection and 2 levels of difference in species richness (circle area). On the left species occurring in the non-protected area are also present in the protected area while the opposite is not true, so the poorer area hosts only a subset of species which is nested in the species composition of the richer area. On the right, species turnover is maximal, with 3 species unique to each protection level and no species in common to both. Increase of species turnover induces a mechanical increase in γ-diversity. At the center, both patterns are present, some species occur in both the non-protected and the protected area (nestedness), but some species are unique to each protection level (turnover). Icons were extracted from https://www.publicdomainpictures.net/and are under the Public Domain Dedication 1.0 license.

species diversity and composition between adjacent (<50 km) Strictly Protected, Restricted, and Non-Protected areas. We define an area as Strictly Protected when human visitation, use, and impacts are strictly controlled and limited, for instance, no-take marine reserves, i.e., International Union for Conservation of Nature (IUCN) categories I and II. In Restricted areas, human activities are controlled but some resource extraction is permitted corresponding to IUCN categories III to VI. We perform 2 complementary statistical analyses at 2 spatial scales to test the relative influence of environmental conditions and protection on species diversities and to shed light on conservation effect, if any.

First, we use distance-based redundancy analyses (dbRDA) at the scale of local surveys (i.e., transects) to disentangle environmental versus protection effects on species composition within each region. Second, we use the Jaccard dissimilarity index (ß-diversity) to determine whether differences in species composition between protection levels are nested in a given region (i.e., species present in less protected areas are also present in more protected areas) or are mainly due to species turnover or replacement along the protection gradient (i.e., species present in protected areas are absent in non-protected areas and vice versa) (Fig 2). Finally, we identify the amount and characteristics of species being unique to one protection level.

## Results

### Both environment conditions and protection level shape species composition

We collected species survey data in 655 protected areas for fish, birds, and plants across the Indo-Pacific reefs, the French Alps, and the United States, respectively (S1 Table). We define a

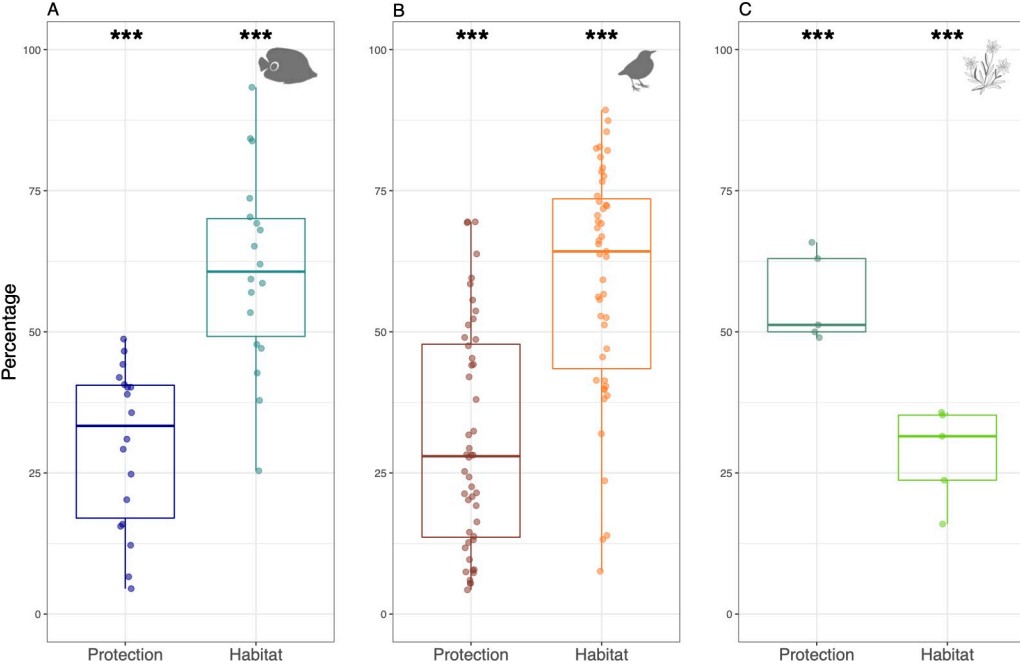

**Fig 2. Result of the partial dbRDAs.** Marginal effect of protection and environmental conditions to observed turnover between Strictly Protected areas surveys and surrounding surveys within a 50-km buffer (from non-protected areas and/ or restricted areas ($n$ = 18, 131, 5) for reef fishes (**A**), birds (**B**), and alpine plants (**C**). Some Strictly Protected areas were removed because environmental conditions were strictly identical between surveys making impossible comparison of the relative contribution of both protection and environmental conditions. Asterisks indicate the Fisher $p$-value, denoting the combined partial effect of protection and environmental conditions for each taxon using permutation ANOVA, i.e., whether or not each environment and protection level had a significant association with observed species turnover. The raw data can be found in https://github.com/LoiseauN/Betadiversity-protected-areas.

region as a Strictly Protected or Restricted area and its surrounding Non-Protected areas within a 50-km buffer, the 3 protection levels being present together in 45 regions. We also extracted 6, 20, and 21 environmental variables recognized as major drivers of species composition for reef fishes, birds, and plants, respectively, on the same surveys (see Materials and methods).

At the scale of individual surveys, we observe a high species turnover within regions with on average 62% (+/−14) for fish, 64% (+/−17) for birds, and 97% (+/− 1) for plants, suggesting a high heterogeneity between surveys. The dbRDA reveal that both environmental conditions and protection significantly explain this turnover in species composition between surveys of the same region (overall F-test = 2.8 ± 1.9, 9.3 ± 5.7, 35.4 ± 9.7, mean adjusted $R^2$ = 0.21 ± 0.1, 0.14 ± 0.06, 0.01 ± 0.05, for fishes, birds, and plants, respectively; S1 Fig). Yet, the explanatory power of dbRDA remains low, particularly for plants, suggesting that, at the scale of surveys, many other microhabitat variations may drive species composition beyond protection and macrohabitat effects. There is a lot of sampling "noise" at this spatial scale.

Partial dbRDA reveal that environmental conditions, after controlling for protection level, significantly explain species turnover between surveys of the same region for the 3 taxa (Fig 2). Symmetrically, protection, after accounting for environmental conditions, also significantly explains species turnover for the 3 taxa (Fig 2). On the balance, environmental conditions have more influence than protection on species turnover between surveys of the same region for birds (70% versus 30%) and fish (69% versus 31%) but this is the opposite for plants (28% versus 56%). These results highlight that differences in environmental conditions cannot explain, alone, the high species turnover between surveys of the same region. Since most of variation in species turnover remains unexplained between surveys within a region, owing to sampling "noise" and unmeasured microhabitat variations, we then pooled surveys by protection level for a given region with the same sampling effort in each protection level (see Materials and methods) for the following analyses. In others words, pooling surveys by protection level may reduce the heterogeneity induced by the sampling performed at very small scale.

## Similar species richness between protection levels

In each region, we randomly sampled 999 times the same number of surveys in Non-Protected areas as in the Strictly Protected or Restricted area to standardize sampling effort and allow species diversity comparisons between pairs of protection levels. We find very low and non-significant differences in the number of bird and reef fish species between the different protection levels within the same region (Fig 3). By contrast, Non-Protected areas in the Alps host significantly more plant species than adjacent Restricted and Strictly Protected areas, while Strictly Protected areas host significantly fewer species than Restricted areas (Fig 3C). We also show that environmental differences between surveys have very low or no significant influence on species richness differences between protection levels across taxa and regions (Fig 4).

## High species turnover between protection levels

Using the same protocol, we show that the level of species dissimilarity between protection levels within the same region is consistently high in the 3 datasets, with a median ß-diversity value ranging between 40% (for birds between Strict Protected and Non-Protected areas) and 80% (for plants between Restricted and Non-Protected areas) (Fig 5) suggesting that adjacent areas (<50 km) with different protection levels host very dissimilar species. Since this result can be due to species richness and occurrence patterns without any effect of protection, we tested whether our observed ß-diversity values among protection levels were larger or smaller than expected under a random assembly model (see Materials and methods). Overall, observed

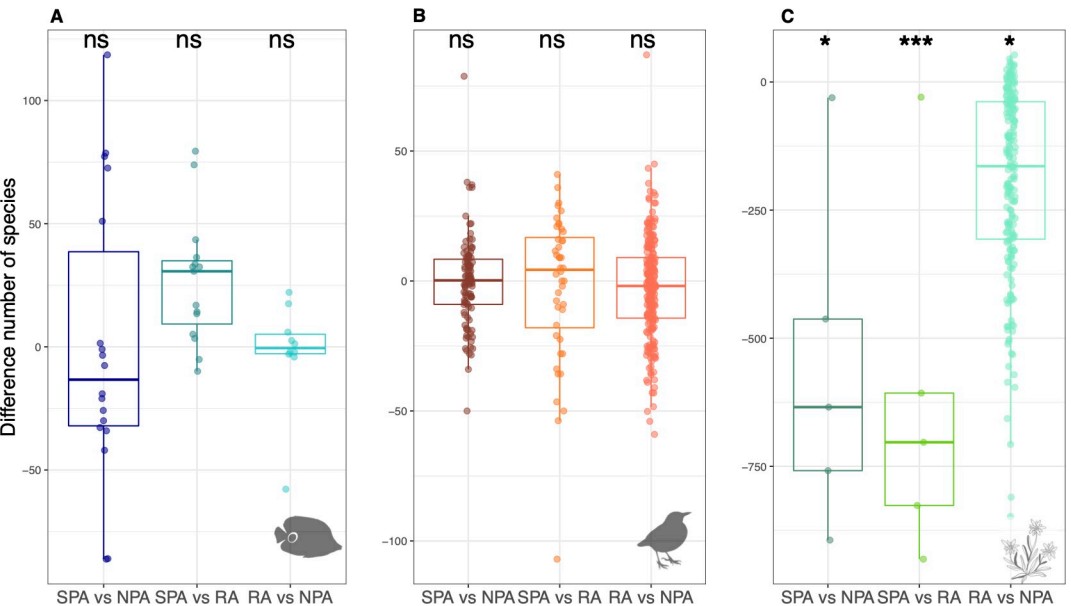

**Fig 3.** Pairwise comparisons of the number of species between SPA, RA, and NPA in surrounding areas within a buffer of 50 km (each dot is a protected area) for reef fishes (**A, D, G**), birds (**B, E, H**), and alpine plants (**C, F, I**). The difference in the number of species, after controlling for the number of surveys inside each protection level, is shown along with the significance of this difference (Wilcoxon): ns for $p > 0.05$, * for $p < 0.05$, ** for $p < 0.01$, and *** for $p < 0.001$. Negative values indicate that areas with the highest protection level host less species than less protected areas. The raw data can be found in https://github.com/LoiseauN/Betadiversity-protected-areas. NPA, Non-Protected Areas; ns, non significant; RA, Restricted Areas; SPA, Strictly Protected Areas.

ß-diversity values between protection levels are higher than expected by chance (combined Fisher $p$-value < 0.001, Fig 5), except for reef fishes between Restricted Protected areas and Non-Protected areas (Fig 5D, S2 Fig). This exception highlights that only Strictly Protected areas play a key and unique role in sustaining regional fish biodiversity.

All datasets combined, the high species dissimilarity between protection levels within a region is primarily due to species turnover (74% ± 20% of the total dissimilarity) and not nestedness (26% ± 20% of the total dissimilarity or ß-diversity). On average, 29% (±14%) of bird, 32% (±8%) of plant, and 43% (±16%) fish species are replaced between adjacent areas under different protection levels, while nestedness represents, on average, between 9% (fish) and 30% (plant) of total species dissimilarity (Fig 5). This proportion of species turnover is also significantly higher than expected at random (combined Fisher $p$-value < 0.001, Fig 5). Only plant species diversity in Strictly Protected areas is strongly nested within plant diversity in nearby Restricted areas (32%, Fig 5F). Like for species richness, this pattern for plants can be explained by the spatial nestedness of Strictly Protected areas within Restricted areas, both nested in Non-Protected areas within regions. The results remain unchanged when smaller (10 km) and larger (100 km) buffers are applied (compared to 50 km; S3 Fig) or when rare species (species with a number of occurrences lower than 3 within a region) are removed (S4 Fig).

We then tested the correlation between the difference in environmental conditions and species turnover for each pair of protection levels within each region for each taxon (Fig 6). Species turnover between Strictly Protected and Non-Protected areas is inconsistently (positive or negative, Fig 6A) and non-significantly related to the difference in environmental conditions for fish ($R^2$ between 0.014 and 0.15, $p$-value > 0.1). For birds and plants, the difference in environmental conditions explains a significant amount of species turnover between Restricted and Non-Protected areas ($R^2$ = 0.24 and 0.40, $p$-value < 0.001, respectively, Fig 6B and 6C) but

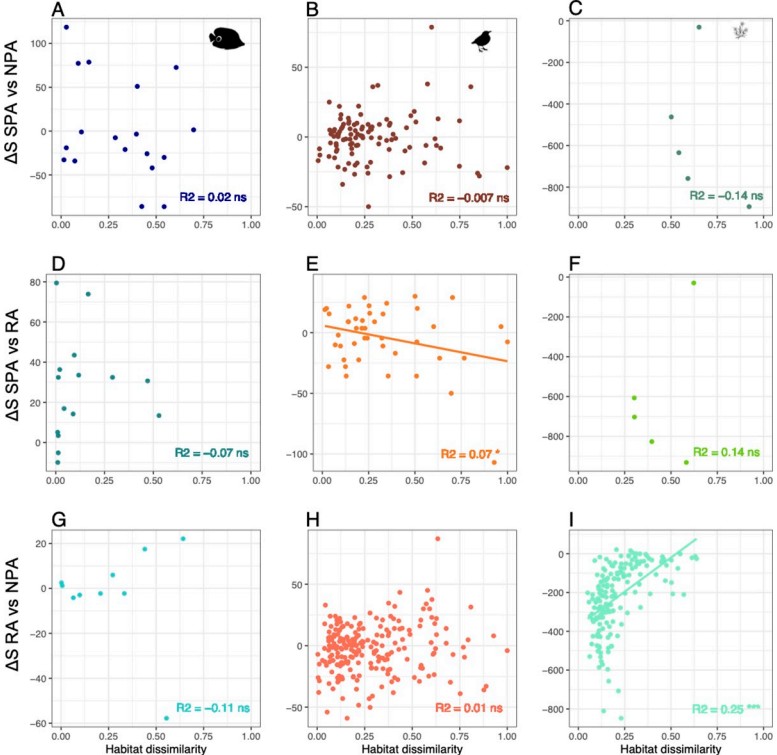

**Fig 4.** Linear relationships between environmental dissimilarity and the difference in species richness (ΔS) for reef fishes (**A, D, G**), birds (**B, E, H**), and alpine plants (**C, F, I**) for pairwise comparisons between SPA, RA, and NPA across regions in a buffer of 50 km (each dot is a protected area). *** $p$-value < 0.001, ** $p$-value < 0.01, * $p$-value < 0.05. The raw data can be found in https://github.com/LoiseauN/Betadiversity-protected-areas. NPA, Non-Protected Areas; ns, non significant; RA, Restricted Areas; SPA, Strictly Protected Areas.

has a weak effect for the other comparisons between protection levels ($R^2$ between 0.09 and 0.3, significant for birds, $p$-value < 0.05, but not for plants $p$-value > 0.1).

## Many species unique to one protection level

When considering only regions with all 3 protection levels (Strictly Protected, Restricted, and Non-Protected areas) in a buffer of 50 km and a standardized subsample (see Materials and methods), we reveal an unexpectedly high proportion of species recorded only in one protection level at the regional scale (Fig 7). For instance, 12%, 13%, and 15% of bird, fish, and plant regional or γ-diversity is only recorded in Non-Protected areas, so absent from nearby Restricted or Strict Protected areas (Fig 7), showing that a non-negligible portion of biodiversity is concentrated in areas where human activities are not buffered through protection. By comparison, 22% of fish species are only recorded in Strictly Protected areas, while this percentage drops to 9% and 4% for birds and plants, respectively. Restricted areas also host a large proportion of unique regional biodiversity ranging from 11% of species for birds to 21% for fishes. Together, Restricted and Strictly Protected areas host, on average, 88%, 87%, and 85% of bird, fish, and plant species regional diversity, respectively.

The proportion of species recorded only in Strictly Protected areas increases when only considering "imperiled" or conservation-dependent species, so a group gathering Critically Endangered, Endangered, Vulnerable, and Near Threatened species, with 58% for fishes, 11% for birds, and 7% for plants (Fig 7D–7F, S4 Fig). This result highlights the capacity of Strictly

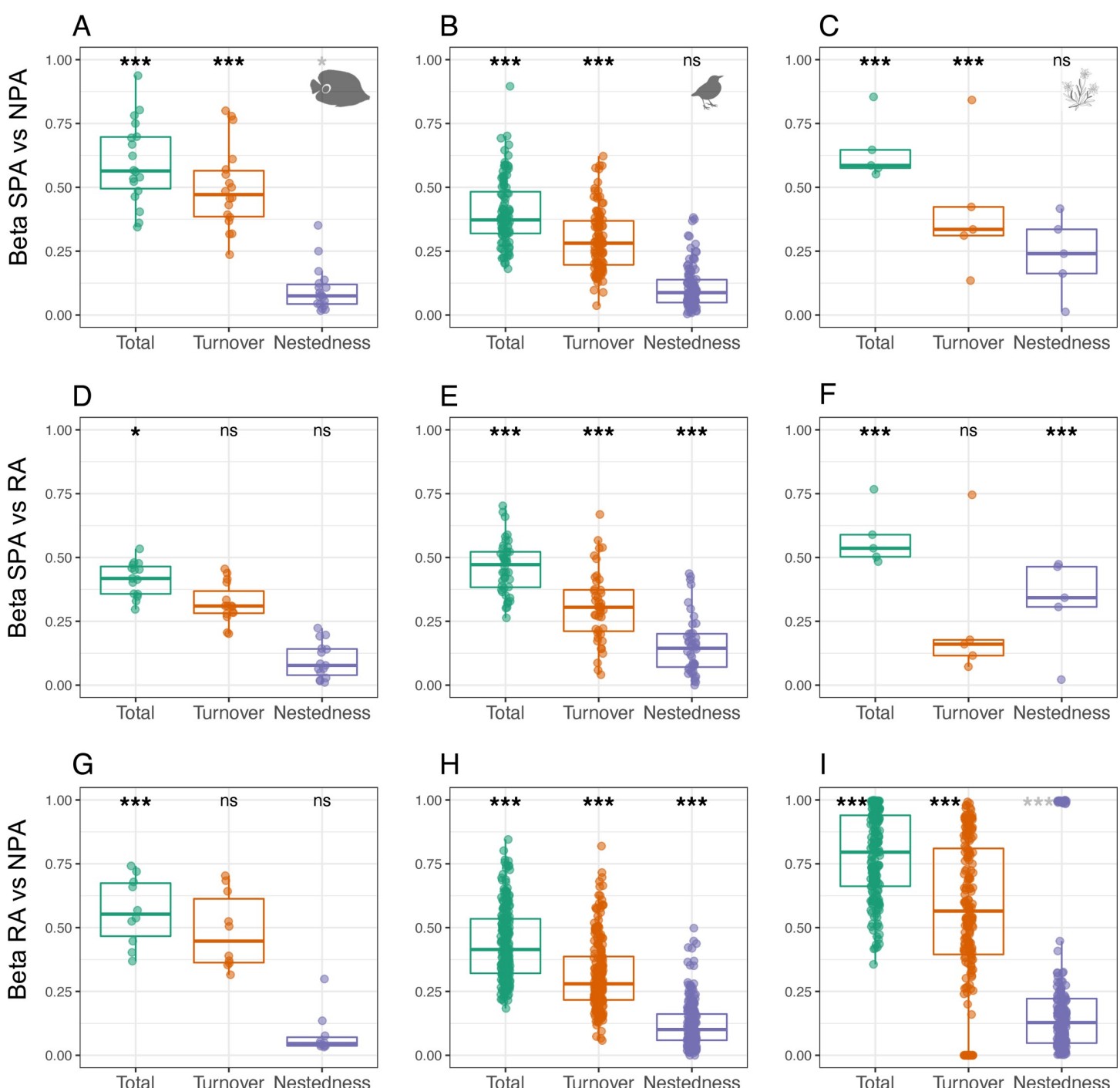

**Fig 5.** Species dissimilarity and its partitioning components of turnover and nestedness between SPA, RA, and NPA in surrounding areas within a buffer of 50 km, after controlling for the number of surveys inside each protection level for reef fishes (**A, D, G**), birds (**B, E, H**), and alpine plants (**C, F, I**). Asterisks indicate the Fisher *p*-value, denoting the probability that the observed combination of dissimilarity values and their partitioning components are different from a random allocation species to surveys; when values of ß-diversity are higher than a random allocation of species to surveys asterisks are in black while for ß-diversity values lower of lower values are in gray (S2 Fig). *** *p*-value < 0.01, ** *p*-value < 0.05, * *p*-value < 0.1. The raw data can be found in https://github.com/LoiseauN/Betadiversity-protected-areas. NPA, Non-Protected Areas; ns, non significant; RA, Restricted Areas; SPA, Strictly Protected Areas.

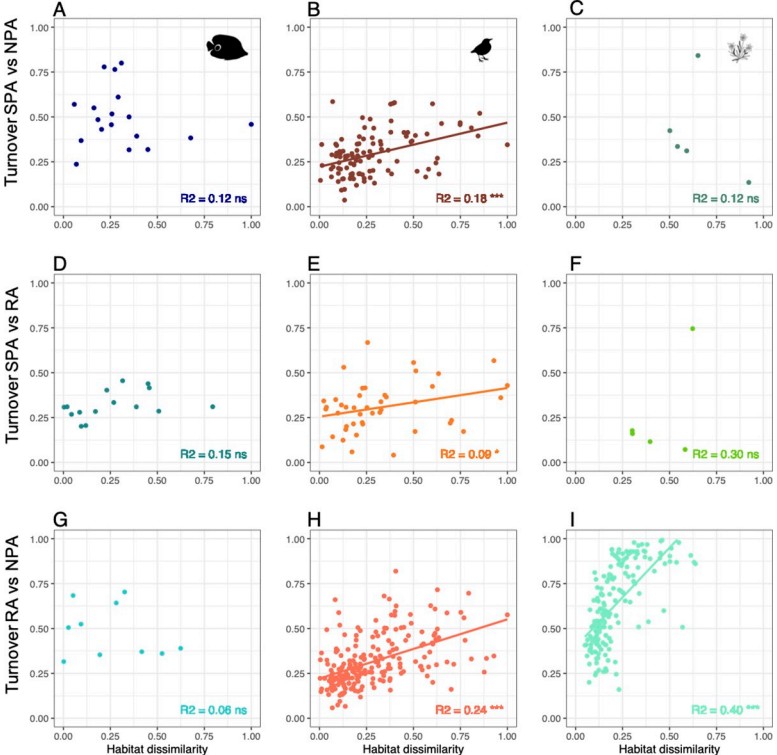

**Fig 6.** Linear relationships between environmental dissimilarity and species turnover for reef fishes (**A, D, G**), birds (**B, E, H**), and alpine plants (**C, F, I**). for pairwise comparisons between SPA, RA, and PA) across regions in a buffer of 50 km (each dot is a protected area). *** *p*-value < 0.001, ** *p*-value < 0.01, * *p*-value < 0.05. The raw data can be found in https://github.com/LoiseauN/Betadiversity-protected-areas. NPA, Non-Protected Areas; ns, non significant; RA, Restricted Areas; SPA, Strictly Protected Areas.

Protected areas to provide refuges for imperiled species. For birds and plants, our results also highlight the critical role of conservation efforts in the protection of imperiled species, since together Restricted and Strictly Protected areas host 94% and 62% of the regional and imperiled species diversity, respectively. Yet, a non-negligible proportion of regional imperiled species (27% for fishes, 6% for birds, and 38% for plants, Fig 7G–7I; S5 Fig) are only recorded in Non-Protected areas, increasing their potential exposure and vulnerability to human activities.

## Discussion

Here, we tested the influence of environmental conditions and protection level on regional biodiversity patterns across 3 taxa and we also used null models and sensibility analyzes to assess the potential effects of species rarity and distribution on those patterns. Since environmental conditions, rarity (patterns remain identical when species with a number of occurrences lower than 3 within a region were removed) or species occurrences cannot fully explain differences in species richness and compositions between protection levels within regions at 2 spatial scales (surveys and areas), a likely explanation for our results is that species have differential responses to direct and indirect human impacts. Humans have transformed ecosystems across more than three-quarters of the terrestrial biosphere [17], while only 13% of oceans globally are classified as wilderness areas [18]. Human activities usually erode ß-diversity by homogenizing landscapes and generating conditions suitable for only a subset of species [19]. In other words, human activities favor the spread of some "winner" species in Non-Protected

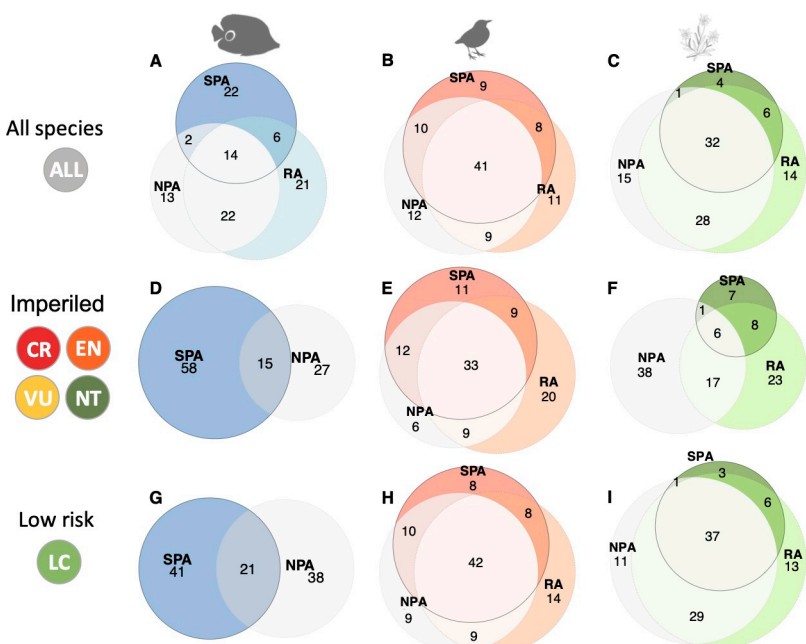

**Fig 7.** Venn diagrams showing the mean percentage of regional gamma species richness shared or unique to the 3 protection levels—SPA, RA, and NPA for reef fishes (**A, D, G**), birds (**B, E, H**), and alpine plants (**C, F, I**). Analyses have been reduced to regions with surveys for the 3 types of protection levels in a buffer of 50 km (6, 43, and 5 strict protected areas, respectively). For fishes, regions with surveys across the 3 types of protection levels in surrounding areas within a buffer of 50 km did not have enough IUCN-assessed species to evaluate the mean percentage of species that are shared and unique. Consequently, for fishes, analyses were performed on the 14 strict protected areas with non-protected area surveys in a buffer of 50 km. CR, EN, VU, and NT species are considered as "Imperiled" (**D–F**) while LC as "Low risk" (**G–I**). The raw data can be found in https://github.com/LoiseauN/Betadiversity-protected-areas. CR, Critically Endangered; EN, Endangered; LC, Least Concern; NPA, Non-Protected Areas; NT, Near Threatened; RA, Restricted Areas; SPA, Strictly Protected Areas; VU, Vulnerable.

Areas to the detriment of "loser" disturbance-sensitive species [20]. For instance, some birds (Galliformes, [16]) or fish predators (sharks, [15]) are known to almost exclusively occur in Strictly Protected areas far from humans. These disturbance-sensitive species may be provided refuge in protected areas, maintaining regional biotic heterogeneity and biodiversity, which are key to ecosystem functioning owing to spatial insurance stabilizing ecological processes across scales [9,10].

Species targeted by hunting and fishing may also adapt their behavior, such as reducing their movement to decrease likelihood of encountering hunters or fishers [21]. Such species may increasingly, sometimes exclusively, occur in Protected areas [22,23], or may become too rare and elusive to be detected in Non-Protected areas [24]. The intensity of human pressure (fishing, hunting, land use, etc.) within Restricted areas could also explain the high number of species not shared with adjacent areas that are Strictly Protected [25–27]. For instance, the presence of livestock and mowing in the French Alps, to maintain the biodiversity of grassland and avoid wood encroachment [26], induces species turnover between Strictly Protected and Restricted areas. This land management is deliberately applied to maintain the biodiversity of alpine grasslands by avoiding succession and dominance by larger plants and forests [26]. Moreover, species abundances differ between Protected and Non-Protected areas [15], altering predator–prey interactions, competition and trophic cascades, and thus species composition [28]. Apex predators are generally the most impacted by human activities and are mostly absent in Non-Protected areas [15]. By favoring the presence of apex predators, Strictly

Protected areas host unique trophic structures and species compositions, different from those observed in Non-Protected or even Restricted areas [29]. Thus, species appear to be "sorting" among different protection levels, and management diversity is needed to maintain larger scale biodiversity.

A non-mutually exclusive but alternative explanation is that protected areas are placed in a way that captures the geographic range of some species but not others [30–32]. Moreover, the simple occurrence within a protected area is insufficient to ensure the long-term persistence of many species, particularly those demanding particular environmental conditions [30,31] or extensive home range [33]. Both mechanisms may lead to the exclusion of some species from Protected areas but also increase the turnover between Protected and Non-Protected areas. In the same way, economic and social dimensions strongly influence the positioning of protected areas. For instance, the most isolated places are easier to protect due to a better social acceptance [34,35]. From this attempt to manage social-ecological systems, it may result that protected species assemblages may not be representative of regional species pools, increasing species dissimilarity between protection levels. At the global scale, protection coverage tends to increase in wilderness areas to the detriment of "Key Biodiversity Areas" [6] so some regional biodiversity hotspots may lack a diversity of management options, notably Strictly Protected areas [36].

Our results reiterate that the number of species living inside versus outside protected areas is a poor indicator of management effectiveness because the identity and composition of species they host can be markedly different while species richness remains similar [37,38]. This contrasts with some studies showing that species richness is 10.6% higher inside protected areas compared to the outside [8]. However, this difference may be due to the difference in spatial extent of pairwise area comparisons and species analyzed between studies since our approach is based on within-region biodiversity patterns using null models and 3 different taxa. Even if our results suggest that species turnover is partly dependent on environmental conditions, the consequence in terms of regional biodiversity conservation remains identical with Strictly Protected, Restricted, and Non-Protected areas not sharing a non-negligible part of species regional pools so being complementary instead of nested. In addition, the influence of environmental conditions is limited by only considering less strictly protected areas within a 50-km buffer (with similar results at 10 and 100 km) around the more protected areas. In other words, changes in macrohabitat (e.g., climate) remain relatively limited over our regional spatial scale. However, from a conservation perspective, expanding protection to environmental conditions that are not already or weakly protected would be highly beneficial for imperiled species [39]. Our data did not allow to determine whether a given protected area was established a priori in specific environmental conditions or if protection induced a differentiation of its environmental conditions a posteriori. For instance, the management of land outside protected areas may favor certain species that are not found within more "natural" environmental conditions like inside protected areas. Finally, future investigation on species identities and functions would reveal mechanisms underlying the high species turnover we observe. Can disparity in trophic level or other functional traits explain changes in species composition between protection levels? What is the contribution of non-native species to the turnover between these management levels?

More generally, our results suggest that human activities do not necessarily reduce $\alpha$-diversity, but instead lead to a shift in species composition, similar to findings assessing biodiversity trends through time [2,39,40]. Human modification of natural environments can even increase local species richness, and thus $\beta$-diversity, because some species can be present in altered conditions only [41,42]. We suggest that, in a world with a predominance of altered landscapes and seascapes, reinforcing Strictly Protected areas with Restricted areas nearby may create a

mosaic of protection levels maximizing the number of species at the regional scale, an important outcome for sustaining ecosystems and their services [14,43,44]. Broadening and diversifying the scope of conservation options to not only focus on local versus global issues but also include regional contexts is critical to prevent future extinctions and to maintain ecosystem multifunctionality under fluctuating and uncertain environmental and human stressors.

## Materials and methods

### Species datasets

**Reef fishes.** We used the global Reef Life Survey (RLS) database [45] and data from coral reef surveys [46]. These reef fish surveys involve underwater visual census (UVC) by SCUBA divers along a 50-m transect line, laid along a depth contour on hard substrata (coral or temperate rocky reef). Observers included highly trained volunteers who produce survey counts statistically indistinguishable from professional researchers [47]. All fish species observed within 5 m of the transect line were recorded as the diver swam slowly along the line. Full details of methods can be found in an online method manual available at www.reeflifesurvey. com [45]. Initially, 39 no-take Strictly Protected areas and 40 Restricted areas were used. However, because habitat was unavailable for most of the reefs, we standardized the habitat of surveys a priori to those performed on reef within the depth range 4 to 20 m. This choice greatly reduced the number of analyzed marine protected areas but minimized any bias due to dissimilarity in habitat. Thus, analyses were reduced to 25 Strictly Protected areas and 19 Restricted areas, plus respective Non-Protected areas. A total of 1,447 species were surveyed along 2,156 transects. We did not compare surveys across the 2 datasets to avoid any methodological bias.

**French alpine plants.** We used a database of vegetation surveys provided by the National Alpine Botanical Conservatory (CBNA) over the French Alps region which covers over 26,000 square kilometers [26]. Database included 42,290 community plots sampled in natural or seminatural areas from 1980 to 2009, encompassing a total of 3,532 plant species. Each plot was approximately 10 m × 10 m and within each sample, species occurrences were recorded. Five Strictly Protected areas and 191 Restricted areas were analyzed.

**Birds.** We use the North American Breeding Bird Survey (BBS), which estimates species bird abundances in 48 US states along 40 km routes. In BBS, yearly, during the breeding season, observers census birds along c. 50 km survey routes with stops separated from 1 km apart and counting all bird species present. We selected the 2,411 routes with surveys from 2010 to 2015. A total of 639 bird species were surveyed. One hundred thirty-two Strictly Protected areas and 283 Restricted areas were analyzed.

**IUCN status.** For each taxon, we grouped species in 3 categories depending on their threat level according to IUCN red list: Critically Endangered (CR), Endangered (EN), Vulnerable (VU), and Near Threatened (NT) as "Imperiled"; Least Concern (LC) species as "Not Threatened"; Data Deficient (DD) species and species without known status as "Not Assessed" species. For fishes, 43 species were classified as "Imperiled," 482 as "Not Threatened," and 922 as "Not Assessed." For birds, 38 species were classified as "Imperiled," 405 as "Not Threatened," and 196 as "Not Assessed." For plants, 280 species were classified as "Imperiled," 1,884 as "Not Threatened," and 1,368 as "Not Assessed." We choose to group "Near Threatened" species with threatened species, i.e., those classified as CR, EN, and VU because we considered that these species should be covered by protected areas networks since being conservation dependent.

### Protection levels

We distinguished 2 different statuses of protected areas: Strictly Protected areas where human activities are prohibited (IUCN I-II) and Restricted areas (IUCN III-VI) where limited

extractive activities are authorized [48]. For birds and plants, we intersected all sites using official coverage of all existing protected areas in US and France (Protected Areas Database of the United States and The World Database on Protected Areas, WDPA). For reef fishes, we used management available data [45,46]. For each survey, we assessed if it was unfished, that is within the borders of a no-take marine reserve; restricted, with active restrictions on particular gear types (for example, bans on the use of nets, spear guns, or traps) or fishing effort (which could have included areas inside marine parks that were not necessarily no-take); or fished, that is, regularly fished without effective restrictions.

## Environmental conditions with habitat and climatic data

**Reefs.**   For global RLS, we used the data from benthic photoquadrats taken along the same transect lines surveyed for fishes to quantify local habitat differences between sites. Photoquadrats were taken vertically downward of the substrate every 2.5 m along each of the same transect lines, and later scored using a grid overlay of 5 points per image, 100 points per transect. Categories of benthic cover scored were from a set of 50 morphological and functional groups of algae and corals. Substratum categories used for analyses here were the sum of all live hard coral categories, macroalgaes, and others. Full details of the photoquadrat method and scoring procedure are already provided [49]. We also included depth of the transect as a local environmental variable. For other data [46], we used habitat (whether the reef is a slope, crest, flat, or back reef/lagoon) and depth recorded along transects. For both databases, we also extracted sea surface temperature and chlorophyll-a concentration using Bio-Oracle at 10 km of resolution average on the period 2000 to 2014 [50]. These variables were all recognized as major drivers of fish diversity on coral reefs [51,52]. As we did not compare surveys across the 2 datasets, differences in environmental datasets should not affect outcomes. Some environmental data were not available for all sites (34% for RLS surveys and 38% for other data [46]). We removed these surveys when testing influence of environmental conditions on diversity.

**French Alps.**   We used the official coverage of land cover in France (Theia Land, 2016, 20 meters of resolutions). This dataset included 17 variables impacting plants: winter agriculture, summer agriculture, hardwood forest, coniferous forest, lawns, woody moors, dense urban, sprawling urban, industrial and commercial areas, roads, mineral surfaces, beaches and dunes, water, glaciers or snow, grasslands, orchards, vineyards. We chose a 1-km buffer as a reasonable range at which principal habitats of recorded species lived and to capture the relevant habitat. We also included average annual precipitation and annual temperature from 1979 to 2013 using Chelsa at 1 km of resolution (http://chelsa-climate.org/).

**US land cover and climate.**   We used the official coverage of land cover in the US (The National Land Cover Database (NLCD, 2016, 30 meters of resolution). NLCD land cover classes (20) were water, perennial ice snow, low intensity residential, high intensity residential, developed/open space intensity, developed/medium intensity, developed high intensity, bare rock/sand/clay, deciduous forest, evergreen forest, mixed forest, shrub/scrub, grasslands/herbaceous, pasture/hay, row crops, woody wetlands, emergent herbaceous wetlands. We created a 19.7-km radius circle (one half the length of a BBS route) around the centroid of each BBS route [53–56] and used this to extract land cover. We chose this radius because it encompassed the entire BBS route, regardless of route path, and a circle because it provided a uniform area and shape around each BBS route. We also included average precipitation and temperature from 1979 to 2013 using Chelsa at 1 km of resolution (http://chelsa-climate.org/). We averaged the climatic data within a 19.7-km radius circle.

All data sources are provided in S2 Table.

## Biodiversity assessments and statistical analyses

**ß-diversity indices.**   When measuring *ß*-diversity, 2 independent patterns may occur, turnover and nestedness [13] (Fig 1). Turnover occurs when species present at one site are absent at another site but are replaced by other species absent from the first. The nestedness component measures differences in richness between assemblages nested to some degree, i.e., species present at one site are absent at another but are not replaced by new species. In order to determine the relative contribution of turnover and nestedness to total *ß*-diversity, we used the additive partitioning of the pairwise Jaccard dissimilarity [13]. This framework teases apart the variation in species composition from species turnover only, which is independent of richness, and from nested patterns (Fig 1).

$$Total\ \beta = Jaccard = \frac{b + c}{a + b + b} = Turnover + Nestedness$$

where

$$Turnover = \frac{2 * \min(b, c)}{a + 2 * min(b, c)}$$

$$Nestedness = \frac{|b - c|}{a + b + c} * \frac{a}{a + 2 * min(b, c)}$$

where a is the number of species present in both sites, b is the number of species present in the first site, but not in the second, and c is the number of species present in the second site, but not in the first.

**Geographical clustering (buffer of fixed radius).**   Beyond the surveys, protected areas (PA) were the spatial units in our study. First, we determined whether a given sample was within a protected area or outside. And then, for those outside, samples within a 50-km buffer of a protected area were designated as Non-Protected area or Restricted area if sample were inside another protected area with a lower level of protection and retained for analyses. The same 50 km buffering approach was applied on restricted areas allowing comparison between restricted and non-protected areas. Because the number of sites sampled outside and inside the protected area was different, we standardized the survey effort by randomly selecting the minimum number of surveys inside and outside, 999 times. With this approach, our design is perfectly balanced in terms of sampled effort between management options. At each iteration, for each site, we took the randomly minimum number of surveys done for all selected sites. For each iteration, we computed *ß*-diversity indices between pooled protected plots and pooled Non-Protected plots and used the average value of the indices. Pooling surveys allows to limit the sampling "noise." To test the robustness of observed patterns, identical clustering methods were applied at 10 and 100 km. We chose 10, 50, and 100 km as a reasonable range to control environmental conditions (habitat and climate).

**Null models.**   We tested whether our observed *ß*-diversity values among levels of protection were larger or smaller than expected under a random assembly model. For each protected area and for each iteration of the bootstrap, we generated 999 random species assemblages using the curveball algorithm using *nullmodel()* function from *vegan v2.4–2* packages [57]. This algorithm maintains species frequency and sample species richness while shuffling species co-occurrences across surveys. In other words, this method maintains row and column totals in a species by survey matrix while shuffling presences within that matrix. Then, we calculated the ß-diversity expected at random. Standardized effect sizes (SES) were calculated using the observed ß-diversity values and the mean and standard deviation of the null distributions. We

plotted (S2 Fig) the average SES per protected area:

$$SES = \frac{observed - mean(null)}{SD(null)}$$

SES values can serve as a measure of departure from a pure null expectation. Values greater than 0 are larger than expected, whereas those smaller than 0 are less than expected. Essentially, departures from 0 indicate non-randomness: Values greater than 1.96 or less than 1.96 are significantly greater or less than expected, at $\alpha = 0.05$. We also derived the *p*-value as the proportion of the null distribution of *ß*-diversity that was more extreme than the observed *ß*-diversity. We combined *p*-values using Fisher methods [58]. *P* values denoted the probability that the observed combination of *ß*-diversity and its partitioning components were different from null expectation.

**Disentangling the effect of protection and environmental conditions.** We performed 2 complementary analyses to tease apart the effects of protection and environmental conditions on species turnover.

**Environmental dissimilarity diversity.** We applied a similar approach as taxonomic beta diversity. At each iteration, for each region, we took the minimum number of surveys done for all selected areas. For each iteration, we computed Euclidean environmental distance using all habitat and climate information between pooled protected surveys and pooled Non-Protected surveys and used the average value of the distance (note that environmental variables were log transformed). With this approach, the area of environmental conditions sampled inside and outside protected areas is perfectly balanced. Then, we tested the link using linear regression between species turnover and average environmental distance for each taxon and each pair of protection levels.

**dbRDA.** At the scale of individual surveys, we performed a complementary explicit test of compositional differences between different environmental conditions and protection levels using distance-based redundancy analyses (dbRDA). dbRDA were centered on Strictly Protected areas. First, we used the extracted environmental conditions data and computed Euclidean environmental distance between each pair of surveys. Then, we performed a PCoA and extracted the first 2 axes. Second, for each Strictly Protected area, we selected all surveys inside and outside the protected area in the buffer of 50 km. Each survey was associated with its protection level and when environmental conditions were available, 2 PCoA axes as explanatory variables. Turnover was calculated for presence–absence matrices. Significance of the models as well as the significance of each axis and of the marginal effect of each variable were tested using ANOVA-like permutation tests with 9,999 permutations. Finally, we computed 2 partial dbRDA using the Jaccard turnover distance to (i) isolate the exclusive effect of protection after accounting for environmental conditions and (ii) isolate the exclusive effect of environmental conditions after accounting for protection.

## Supporting information

**S1 Table. Information on sampling size for Strictly Protected Areas (SPA), Restricted Areas (RA), and Non-protected Areas (NPA) for birds, reef fishes, and alpine plants.**
(PDF)

**S2 Table. Names and sources of datasets used in the present study.**
(PDF)

**S1 Fig. Result of the partial dbRDAs.** Rsquare of the marginal effect of protection and habitat to observed turnover between Strictly Protected areas surveys and surrounding surveys within

a 50-km buffer (from non-protected areas and/or restricted areas ($n$ = 18, 131, 5) for reef fishes, birds, and alpine plants. Some Strictly Protected was removed because habitat was strictly identical between surveys making impossible comparison of the relative contribution of both protection and habitat. The raw data can be found in https://github.com/LoiseauN/Betadiversity-protected-areas.
(PDF)

**S2 Fig. Boxplot showing SES values of multiple-site beta diversities (green) and their partitioning components of turnover (orange) and nestedness-resultant (blue) computed between SPA, RA, and NPA for reef fishes, birds, and plants.** Bold black dotted line indicate 0 SES, for random values. Black dotted lines indicate the α = 0.05 threshold of 1.96 SES for significantly nonrandom values. Each data point represents a comparison between protected area and its proximate less protected area (in a buffer of 50 km, 100 km, and 10 km). The raw data can be found in https://github.com/LoiseauN/Betadiversity-protected-areas. NPA, Non-Protected Area; RA, Restricted Area; SES, standardized effect size; SPA, Strict Protected Area.
(PDF)

**S3 Fig. Boxplots showing the multiple-site beta diversities (green) and their partitioning components of turnover (orange) and nestedness-resultant (blue) computed between SPA, RA, and NPA in a buffer of 100 km and 10 km.** The raw data can be found in https://github.com/LoiseauN/Betadiversity-protected-areas. NPA, Non-Protected Area; RA, Restricted Area; SPA, Strict Protected Area.
(PDF)

**S4 Fig. Boxplots showing the multiple-site beta diversities (green) and their partitioning components of turnover (orange) and nestedness-resultant (blue) computed between SPA, RA, and NPA when rare species (species with a number of occurrences lower than 3) are removed.** The raw data can be found in https://github.com/LoiseauN/Betadiversity-protected-areas. NPA, Non-Protected Area; RA, Restricted Area; SPA, Strict Protected Area.
(PDF)

**S5 Fig. Probability of presence of reef fishes, birds, and alpine plants species for the 3 management types—SPA, RA, and NPA without any geographical buffer and control of sampling effort.** CR, EN, VU, and NT species as "Imperiled"; LC as "Low risk." Stars indicate significance of Tukey post hoc test computed after the analysis of variance. *** $p$-value $< 0.01$, ** $p$-value $< 0.05$, * $p$-value $< 0.1$. The raw data can be found in https://github.com/LoiseauN/Betadiversity-protected-areas. CR, Critically Endangered; EN, Endangered; LC, Least Concern; NPA, Non-Protected Areas; NT, Near Threatened; RA, Restricted Areas; SPA, Strictly Protected Areas; VU, Vulnerable.
(PDF)

## Acknowledgments

We are grateful to Dr. John R. Sauer for his comments on BBS surveys. We acknowledge the RLS divers who with the reef fish data collection, and A. Cooper, J. Berkhout, E. Clausius and E. Oh for RLS data management.

## Author Contributions

**Conceptualization:** Nicolas Loiseau, Wilfried Thuiller, Rick D. Stuart-Smith, Vincent Devictor, Graham J. Edgar, Laure Velez, Joshua E. Cinner, Nicholas A. J. Graham, Julien Renaud, Andrew S. Hoey, Stephanie Manel, David Mouillot.

**Data curation:** Nicolas Loiseau, Rick D. Stuart-Smith, Graham J. Edgar, Laure Velez, Julien Renaud.

**Formal analysis:** Nicolas Loiseau.

**Funding acquisition:** Wilfried Thuiller, Stephanie Manel.

**Investigation:** Nicolas Loiseau, Wilfried Thuiller, David Mouillot.

**Methodology:** Nicolas Loiseau, Wilfried Thuiller, David Mouillot.

**Resources:** Wilfried Thuiller, David Mouillot.

**Software:** Nicolas Loiseau.

**Validation:** Wilfried Thuiller, David Mouillot.

**Visualization:** Nicolas Loiseau.

**Writing – original draft:** Nicolas Loiseau, David Mouillot.

**Writing – review & editing:** Wilfried Thuiller, Rick D. Stuart-Smith, Vincent Devictor, Graham J. Edgar, Joshua E. Cinner, Nicholas A. J. Graham, Andrew S. Hoey, Stephanie Manel.

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
