## [Editor Report · Decision Letter 0]

21 Jan 2021

Dear Nicolas, 

Thank you for submitting your revised manuscript entitled "Maximising regional biodiversity requires a mosaic of protection levels" for consideration as a Research Article by PLOS Biology.

Your revisions have now been evaluated by the PLOS Biology editorial staff, and I'm writing to let you know that we would like to send your submission out for re-review.

Please re-submit your manuscript within two working days, i.e. by Jan 25 2021 11:59PM.

Kind regards,

Roli

Senior Editor

PLOS Biology

---

## [Decision Letter · Decision Letter 1]

4 Mar 2021

Dear Dr Loiseau,

Thank you for submitting your revised Research Article entitled "Maximising regional biodiversity requires a mosaic of protection levels" for publication in PLOS Biology. I have now obtained advice from two of the original reviewers and have discussed their comments with the Academic Editor. 

Based on the reviews, we will probably accept this manuscript for publication, provided you satisfactorily address the remaining points raised by the reviewers. Please also make sure to address the following data and other policy-related requests.

IMPORTANT:

a) Please address the remaining concerns raised by reviewers #1 and #3.

b) Note that reviewer #1 takes issue with your current title. However you address this concern, please ensure that the revised title is informative and accessible to our broad readership.

c) Please also address the points mentioned by the Academic Editor at the foot of the email. Unfortunately reviewer#2 was unable to re-review, so I'd asked the Academic Editor to assess your responses to rev #2's concerns, in addition to those to his/her own comments.

d) Please attend to my Data Policy requests further down, namely to supply the numerical values underlying Figs 2ABC, 3ABC, 4ABCDEFGHI, 5ABCDEFGHI, 6ABCDEFGHI, S1, S2, S3, S4, S5, and to cite the location of these data in all respective Figure legends.

We expect to receive your revised manuscript within two weeks. 

*Published Peer Review History*

*Early Version*

Sincerely,

Roli Roberts

Senior Editor,

rroberts@plos.org,

PLOS Biology

DATA POLICY:

Regardless of the method selected, please ensure that you provide the individual numerical values that underlie the summary data displayed in the following figure panels as they are essential for readers to assess your analysis and to reproduce it: Figs 2ABC, 3ABC, 4ABCDEFGHI, 5ABCDEFGHI, 6ABCDEFGHI, S1, S2, S3, S4, S5. NOTE: the numerical data provided should include all replicates AND the way in which the plotted mean and errors were derived (it should not present only the mean/average values).

DATA NOT SHOWN?

- Please note that per journal policy, we do not allow the mention of "data not sown", "personal communication", "manuscript in preparation" or other references to data that is not publicly available or contained within this manuscript. Please either remove mention of these data or add figures presenting the results and the data underlying the figure(s).

REVIEWERS' COMMENTS:

Reviewer #1:

I was reviewer 1 in the previous round. I'm largely satisfied with the responses of the reviewers to my concerns, especially regarding beta-diversity calculations. I'm a bit less certain why the authors have chosen to ignore (i.e., no measure/report) some indication of larger scale (gamma) diversity. But I will not quibble too much. My remaining concern, then, lies in the title. The authors are discussing 'maximizing' regional biodiversity, but not directly measuring it. I understand that the Baselga measures get you part way there, but why not just measure regional diversity itself? Otherwise, the title seems misleading, as well as bits of the abstract. Scale is a real thing in biodiversity studies, and beta diversity, at least as measured here, obscures it as much as helps it, whereas measuring richness and other measures at multiple scales (ie., SARs, rarefaction curves or the like) allow you to directly look at this.

Reviewer #3:

the authors have done a very thorough job of responding to the comments made by myself and the other reviewers. In particular, they have added extra analyses to parse out the contributions of habitat versus protection differences on observed turnover in species composition. I have a few suggestions for changes to the text that I hope may further improve the paper.

My main remaining concern is around the conservation recommendations that emerge from the findings. On lines 253-256, the authors contend that consequences are identical whether or not habitat plays a substantial role in driving observed differences in species composition. I don't agree. If it is the climate component of habitat differences that drive the differences in species composition, then it does not necessarily follow that having a mosaic of different protection levels will benefit biodiversity. Expanding protection to the climates (and thus species) that are not already protected would be most beneficial. Along the same lines, the authors respond that considering the identity of the species found in different protection levels (including whether or not they are invasive species) is beyond the scope of the paper. That is perfectly reasonable, but it would be good to add some discussion of the conservation implications of species identity. Different conservation perspectives will place different values on different species, which is important for informing management recommendations.

Related to my previous comment, I am not sure about referring to habitat dissimilarity, since the measures of 'habitat' include climate as well as land cover, reef structure etc. The reader has to get a long way through the paper (until the Methods section at the end) before realising that the methods controlled for climatic differences. I wonder if a more inclusive label could be used? Or at least, it would be good to state early on that climate was included in the analysis.

The supplementary analysis of the standardised effect sizes highlights that strictly protected areas are important for sustaining reef fish biodiversity. But I don't think this important result is discussed in the main text.

Specific comments:

Lines 65-68: This sentence doesn't follow neatly from the previous one.

Lines 80-81: It is odd to talk of a 'pure' conservation effect. In reality, outcomes will depend on a complex mix of abiotic factors, natural biotic processes, and human activities.

Line 83: This would be a good point at which to state that the 'habitat' effects also included measures of climate.

Lines 101-102: The turnover numbers presented here won't be intuitive for readers unfamiliar with the specific indices used. Could the statistics be restated in a way that is easier to interpret?

Line 107: The 'sampling effect' hasn't been introduced yet.

Line 119: "owing to"

Lines 119-121: The reason for pooling surveys within protection levels needs unpacking in more detail here.

Line 126: "as in" not "than in"

Line 132: In what sense is this "symmetrically"?

Line 132: Clarify whether these "habitat differences" also include the climate variables.

Line 141-143: The rationale for the null modelling approach introduced here needs more explanation.

Line 148-150: Do the statistics presented in this sentence reflect the results for all datasets combined?

Line 155: Should this be "strongly nested within plant diversity"?

Line 162: Again, clarify whether the "habitat difference" here included the effects of the climate variables.

Lines 168-169: For birds at least, Figure 6 seems to show a significant positive relationship for all three comparisons.

Line 197: It hasn't been very clear yet how the methods accounted for the effects of species rarity.

Line 243: "Key Biodiversity Areas"

Line 259-260: I am not convinced by the argument that "change in macro-habitat (e.g. climate) remains relatively limited over our regional spatial scale". In many regions, climate can change markedly over 50 km. For example, in areas of high topographic complexity (such as the Alps!).

Line 302: "The database included" not "Database include"

Lines 312-313: For the analysis of the bird dataset, presumably the BBS routes had to fall entirely within a given protection level to be included in the analysis?

Line 319-323: These estimates of 8-13% species considered imperiled are much lower than the %s reported in the IPBES global assessment. Is that just to do with the regions considered?

Line 330: Give a reference where the criteria for distinguishing IUCN categories are given in full.

Lines 330-331: Specify here again that the 'Restricted Areas' correspond with IUCN categories III-VI.

Lines 333-334: Say a little more about the origin of these "local data".

Line 368: Is 'average precipitation' the average of total annual precipitation, and 'temperature' mean annual temperature?

Lines 369-370: What choice is highly conservative? In what sense is it highly conservative?

Line 373: 'US land cover' is not an informative title, given that both land cover and climate data were included.

Lines 383-384: How was the 1 km spatial resolution of the climate data reconciled with the fact that the BBS routes spanned 40 km?

Lines 410-414: Was it possible to have a comparison consisting of just restricted areas and non-protected areas? Was the same 50-km buffering approach applied?

Lines 419-420: The reason for this pooling is never made very clear.

Lines 422-423: Wasn't the buffering approach intended to control, as much as possible, the climatic conditions surveyed? Potential spillover of species is a separate issue.

Line 452: "applied a similar approach as taxonomic beta diversity".

Line 454: Specify here whether the "habitat information" included the climate data.

Lines 466-467: Was the PCoA based on the habitat information? Did this include the climate data?

Line 469: Under what circumstances were the PCoA axes not available?

Lines 472-474: More details are needed here on the computation of the partial dbRDA.

Lines 676-677: I.e. whether or not each of habitat and protection had a significant association with observed turnover?

Lines 699-700: "a random allocation of species to surveys"

Figure 5: It is odd to use "turnover" as the y-axis label, when turnover is just one of the components of beta diversity considered. Perhaps use "dissimilarity".

Supplementary Information, Figure S3 legend: Does "at 100km and 10km" refer to the buffers of 100 km and 10 km?

Figure S5 legend: "significance" not "significativity"

COMMENTS FROM THE ACADEMIC EDITOR [lightly edited]

The response to my comments seem fine, but the addition of "circle area" to the Fig. 1 caption doesn't fit for the Nestedness case, as the circle with three black birds on the far left (Nestedness = 1) is smaller than the other two circles that also have 3 species. Seems trivial but for these types of schematics to work, they have to be very clear.

As for reviewer #2's comments, I agree the major ones mostly overlap with the other Reviewers, who seem satisfied. The one exception is the comment about L154-155 in the original MS with response R2.5 from the authors. I don't understand what the authors have done, if anything, to address this comment.

---

## [Editor Report · Decision Letter 2]

18 Mar 2021

Dear Dr Loiseau,

On behalf of my colleagues and the Academic Editor, Andrew Tanentzap, I'm pleased to say that we can in principle offer to publish your Research Article "Maximising regional biodiversity requires a mosaic of protection levels" in PLOS Biology, provided you address any remaining formatting and reporting issues. These will be detailed in an email that will follow this letter and that you will usually receive within 2-3 business days, during which time no action is required from you. Please note that we will not be able to formally accept your manuscript and schedule it for publication until you have made the required changes.

PRESS: We frequently collaborate with press offices. If your institution or institutions have a press office, please notify them about your upcoming paper at this point, to enable them to help maximise its impact. If the press office is planning to promote your findings, we would be grateful if they could coordinate with biologypress@plos.org. If you have not yet opted out of the early version process, we ask that you notify us immediately of any press plans so that we may do so on your behalf.

Thank you again for supporting Open Access publishing. We look forward to publishing your paper in PLOS Biology. 

Sincerely, 

Roli Roberts

Roland G Roberts, PhD 

Senior Editor 

PLOS Biology